# Multi-Omics Analysis of a Chromosome Segment Substitution Line Reveals a New Regulation Network for Soybean Seed Storage Profile

**DOI:** 10.3390/ijms25115614

**Published:** 2024-05-21

**Authors:** Cholnam Jong, Zhenhai Yu, Yu Zhang, Kyongho Choe, Songrok Uh, Kibong Kim, Chol Jong, Jinmyong Cha, Myongguk Kim, Yunchol Kim, Xue Han, Mingliang Yang, Chang Xu, Limin Hu, Qingshan Chen, Chunyan Liu, Zhaoming Qi

**Affiliations:** 1National Key Laboratory of Smart Farm Technology and System, Key Laboratory of Soybean Biology in Chinese Ministry of Education, College of Agriculture, Northeast Agricultural University, Harbin 150030, China; 13030046075@163.com (C.J.); yuzhenhai1900@126.com (Z.Y.); zhangyu_05101999@163.com (Y.Z.); cuijinghao1981@163.com (K.C.); zheng@neau.edu.cn (S.U.); kgb19841123@163.com (K.K.); jc0087117@163.com (C.J.); chejinming1983@163.com (J.C.); k15124579051@163.com (M.K.); 13134518306@163.com (Y.K.); hanxue1860@126.com (X.H.); yml5418@126.com (M.Y.); xuchang@neau.edu.cn (C.X.); hulimin@neau.edu.cn (L.H.); cyliucn@126.com (C.L.); 2Heilongjiang Green Food Science Research Institute, Harbin 150000, China

**Keywords:** multi-omics analysis, CSSLs, seed fatty acid, seed storage protein

## Abstract

Soybean, a major source of oil and protein, has seen an annual increase in consumption when used in soybean-derived products and the broadening of its cultivation range. The demand for soybean necessitates a better understanding of the regulatory networks driving storage protein accumulation and oil biosynthesis to broaden its positive impact on human health. In this study, we selected a chromosome segment substitution line (CSSL) with high protein and low oil contents to investigate the underlying effect of donor introgression on seed storage through multi-omics analysis. In total, 1479 differentially expressed genes (DEGs), 82 differentially expressed proteins (DEPs), and 34 differentially expressed metabolites (DEMs) were identified in the CSSL compared to the recurrent parent. Based on Gene Ontology (GO) term analysis and the Kyoto Encyclopedia of Genes and Genomes enrichment (KEGG), integrated analysis indicated that 31 DEGs, 24 DEPs, and 13 DEMs were related to seed storage functionality. Integrated analysis further showed a significant decrease in the contents of the seed storage lipids LysoPG 16:0 and LysoPC 18:4 as well as an increase in the contents of organic acids such as L-malic acid. Taken together, these results offer new insights into the molecular mechanisms of seed storage and provide guidance for the molecular breeding of new favorable soybean varieties.

## 1. Introduction

Soybean, one of the major cultivated crops worldwide, is sought after as a rich source of oil and protein. Because soybean is an important source of edible oil and other food products, its ever-increasing demand needs to be satisfied and sustained by developing new soybean varieties with favorable qualities. Soybean seed development can be divided into early stage (EM stage), middle stage (MM stage), late stage (LM stage), and dry seed stage (DS stage). During the EM period, lipid droplets and storage proteins have not yet begun to form. During the MM period, lipid droplets form, but storage proteins have not yet formed. During the LM period, lipid droplets and storage proteins exist. During the DS period, lipid droplets and storage proteins accumulate. At this time, protein and lipid accumulation in soybean seeds reaches their maximum.

Fatty acids (FAs) and seed storage proteins (SSPs) are the main storage compounds in soybean seed and a thorough understanding of the molecular mechanisms for their synthesis and accumulation is a prerequisite for soybean improvement [1,2]. Unsaturated FAs in soybean, such as linolenic acid, play important roles in immune system regulation, neurotransmission, and cholesterol metabolism as well as blood clotting regulation [3]. FAs in soybean oil consist of five main species, namely palmitic acid (C16:0), stearic acid (C18:0), oleic acid (C18:1), linoleic acid (C18:2), and linolenic acid (C18:3), with a respective content ratio of about 10:4:18:55:13 [4,5,6]. Generally, free FAs synthesized from plastids are transported to the endoplasmic reticulum, where triacylglycerols (TAGs) are produced through a co-expression network of several proteins and then stored in lipid droplets (LDs). FA synthesis in soybean has been well characterized, and functional analysis of the key genes involved in FA synthesis and TAG accumulation has provided copious information for developing new soybean varieties with high oil content and optimal FA composition [7,8,9,10].

Soybean proteins are widely applied in the food industry, and they can lower serum cholesterol and plasma TAG levels in the human body. In general, the total protein content in soybean seed ranges from 37% to 42% on a dry weight basis, with β-conglycinin and 11S glycinin accounting for 70% to 80% of the total seed protein [11,12]. The first sequenced soybean genome was v. Williams 82, a cultivar developed in America in the 1980s [13]. Many studies have focused on increasing the nutritional value of soybean by optimizing the amino acid composition and increasing the total protein content in seeds. For example, a previous study compared the expression level of asparagine synthetase (AS) among five Chinese soybean cultivars and revealed an association between the AS expression level and the seed protein content [14]. A mutagenesis of nine seed storage protein genes by CRISPR/Cas9 caused a change in protein content in soybean hairy root [15]. Overexpression of the species-specific Qua-Quine Starch (QQS) protein from *Arabidopsis* in soybean led to a 13% decrease in seed oil content and an 18% increase in seed protein content [16]. In addition, overexpression of the transcription factor GmNF-YC4-2 (Glyma.04G196200), which interacts with the QQS protein, decreased leaf starch content and increased seed protein content [17]. Furthermore, a two-base pair deletion (CC) in GmSWEET39 was associated with changes in seed protein and oil content [2].

Transcriptomics, proteomics, and metabolomics have been widely adopted in plant breeding and food industries with singular and combined (multi-omics) applications, providing broader insights into the molecular mechanisms regulating important biological processes, including seed storage processes. For example, researchers have identified 20,039 genes and 471 proteins through integrated transcriptomic and proteomic analyses of Coix seeds and have established a co-expression network for nutrient metabolism and accumulation, including primary metabolism (starch, sucrose, lipid, and amino acid metabolism) and flavonoid biosynthesis [18]. Integrated transcriptomic and proteomic analyses in soybean (*Glycine max* (L.) Merr.) revealed differences in the expression levels of genes and metabolites participating in different metabolic pathways in soybean seed, potentially leading to the discovery of new candidate genes that can be used for seed quality improvement [19]. Multi-omics has also been widely applied in various studies to reveal the molecular mechanisms of biological responses to biotic and abiotic stresses [20,21,22].

Since their first reported application in tomato in 1994, CSSLs have been considered a valuable material for crop breeding and genetic analysis. Each CSSL line carries a single or a few chromosomal segments from a donor in the genetic background of a recurrent parent [23]. In 2013, the development of CSSLs was first reported in soybean, and since then, they have been utilized for the identification and functional analysis of favorable genes related to agronomic traits, including seed size, yield, and quality [24,25,26,27,28]. However, few studies have addressed the regulatory network driving the synthesis and accumulation of seed storage compounds in soybean CSSLs. Previously, we constructed a CSSL population of 194 lines derived from SN14 soybean (receptor plant or recurrent parent) introgressed with chromosome segments of wild soybean ZYD00006 (donor plant) [28]. In this study, one CSSL line with high protein content and low oil content was selected from this CSSL population through the analysis of its seed FA and SSP contents. In addition, we conducted a multi-omics analysis, hoping to screen out new candidate genes and metabolites involved in the accumulation of FA and SSP, and provide new ideas for the accumulation of soybean nutrients.

## 2. Results and Analysis

### 2.1. Selection of a CSSL with High Protein Content and Low FA Content

We identified a line, R122, which showed a higher protein content and lower FA content compared with SN14 from our previously published CSSL population (Figure 1).

The palmitic acid, stearic acid, oleic acid, linoleic acid, and linolenic acid contents of R122 were 0.80 to 0.82%, 0.08 to 0.20%, 1.89 to 1.93%, 1.79 to 1.89%, and 0.09 to 0.18% lower than those in SN14, respectively, and the total FA content was 4.65% and 5.02% lower than that in SN14 in 2019 and 2020, respectively (Figure 1A,B). In contrast to the FA content, the total protein content was 2.12% and 1.93% higher than that in SN14 in 2019 and 2020, respectively (Figure 1D). Total protein was extracted and quantified by gradient SDS-PAGE electrophoresis, and the levels of 11S and 7S in R122 did appear stronger than those in the recurrent parent (Figure 1C). Whole-genome sequencing revealed that ZYD00006 chromosomal segments were introduced into five chromosomes, which explained the differences in seed storage composition between R122 and SN14 (Appendix A).

### 2.2. RNA-Seq, Proteomics, and Metabolomics Analyses of R122 and SN14

We performed RNA-seq, proteomics, and metabolomics analyses using dried seeds of R122 and SN14 to further investigate the molecular mechanism that underlies the different seed storage profiles of R122 and SN14. The total clean reads obtained from RNA-seq analysis were approximately 299 M, and the mapping ratio ranged from 94.44% to 95.40% (Appendix A). Analysis of the total reads identified 35,117 expressed genes (Appendix A), and 1479 DEGs (log2 FC >1 and <−1 [*p* < 0.05]) were detected. Among these DEGs, 1167 were up-regulated and 312 were down-regulated in R122 (Appendix A). DEGs were annotated with 19 GO terms (*p* < 0.05)—8 biological process terms, 3 cellular component terms, and 8 molecular function terms—including seed-storage-related terms, such as nutrient reservoir activity (GO: 0045735), cysteine-type endopeptidase inhibitor activity (GO: 0004869), L-phenylalanine biosynthetic processes (GO: 0009094), lipid biosynthetic processes (GO: 0008610), the regulation of macromolecule metabolic processes (GO: 0060255), and amino acid transmembrane transporter activity (GO: 0015171) (Figure 2A). KEGG annotation of the DEGs revealed the enrichment of pathways related to seed oil and protein content, including cysteine and methionine metabolism (ko00270); valine, leucine, and isoleucine biosynthesis (ko00290); lysine biosynthesis (ko00300); glycolysis/gluconeogenesis (ko00010); and FA degradation (ko00071) (Figure 2B). Of the DEGs identified, 63 resided within chromosomal substitution regions. The NCBI and SoyBase databases were used to annotate the putative homologous genes within these regions (Appendix A).

To determine if any expression patterns of DEGs correlated with changes in protein profiles, we performed proteomics analysis on the seeds of SN14 and R122. We identified 18,055 peptides and 3415 proteins from 80,732 peptide spectrum matches (Appendix A), of which 82 were DEPs (FC >1.2 or <0.83 [Student’s *t*-test: *p* < 0.05]) (Appendix A). We performed GO term analysis, and the DEPs were classified into 10 biological process terms, 5 cellular component terms, and 10 molecular function terms, including seed maturation (GO: 0010431), fatty acid binding (GO: 0005504), endopeptidase inhibitor activity (GO: 0004866), and nascent polypeptide-associated complex (GO: 0005854) (Figure 3A).

We then performed KEGG annotation and observed enrichment in pathways related to seed storage, including the biosynthesis of amino acids (gmx01230); cysteine and methionine metabolism (gmx00270); phenylalanine, tyrosine, and tryptophan biosynthesis (gmx00400); protein processing in the endoplasmic reticulum (gmx04141); and ubiquitin-mediated proteolysis (gmx04120) (Figure 3B).

Up-regulated pathways included linoleic acid metabolism (gmx00591); amino acid biosynthesis (gmx01230); phenylalanine, tyrosine, and tryptophan biosynthesis (gmx00400); and ubiquitin-mediated proteolysis (gmx04120) (Appendix A). Among the 82 DEPs, 5 proteins were most abundant: C6T129 and I1K6H0 are involved in ubiquitin-mediated post-translational modifications (>1.6 FC), and I1J702 is involved in protein processing in the endoplasmic reticulum, I1KP94 (glutathione peroxidase), and A0A0R0KI45 (Glyma.03G124300: uncharacterized protein) [both >1.5 FC] (Appendix A). Additionally, we identified three up-regulated DEPs in the chromosomal substitution region of R122, including I1JPV1 (TRASH domain-containing protein), I1KXH0 (spermidine synthase 1), and A0A0R0KV40 (a lipoxygenase). A total of 52 DEPs representing down-regulated pathways were identified, including oxidative phosphorylation (gmx00190) and amino sugar and nucleotide sugar metabolism (gmx00520) (Appendix A). The five DEPs with the largest decrease in abundance included A1KR24 (dehydrin), Q39842 (cysteine proteinase inhibitor), K7K348 (involved in carbon and nitrogen metabolism), C7S8D1 (germin-like protein), and A0A6S5ZYU0 (seipin 1A) (Appendix A). Interestingly, I1KUN9 (transmembrane 9 superfamily member) in the chromosomal substituted regions of R122 was significantly down-regulated. In total, four DEPs from the chromosomal substituted regions of R122 which were responsible for seed storage, showed a significant difference in expression level, which may be functionally relevant for the FA and SSP profile differences observed between SN14 and R122 [29,30].

We then performed metabolic analysis to determine if the DEGs and DEPs identified corresponded to changes in the metabolic profile of R122 seeds. Among the 574 identified metabolites (Appendix A), 34 DEMs with FC >2 or <0.5 and VIP > 1 were considered for comparative analysis. These DEMs included 13 amino acids and derivatives, 3 lipids, 12 organic acids, 1 nucleotide and derivative, and 5 saccharides and alcohols (Appendix A). We then performed KEGG annotation and observed the enrichment of several pathways related to seed storage, including alanine, aspartate, and glutamate metabolism (ko00250); arginine and proline metabolism (ko00330); lysine biosynthesis (ko00300); glycolipid metabolism (ko00561); and carbon fixation in photosynthetic organisms (ko00710) (Figure 4).

For the specific DEMs identified, we found that the abundance of amino acid derivatives, including N-acetyl-L-glycine, D-proline betaine, S-(2-carboxypropyl) cysteine, L-lysine-butanoic acid, and N-carbamoyl-L-aspartate, increased in R122, whereas metabolites from other pathways, including L-arginine, LysoPG 16:0, and LysoPC 18:4, decreased in R122. Other DEMs, such as L-malic acid (involved in carbon fixation in photosynthetic organisms) and γ-aminobutyric acid (involved in amino acid synthesis), were 2.12- and 2.95-fold more abundant in R122 than in SN14, respectively. In addition, saccharides, such as acetyl-D-mannosamine and glucose-1-phosphate, and alcohols, such as ribitol, showed increased abundance in R122 (Appendix A).

Overall, multi-omics analyses revealed that R122 possesses a unique molecular fingerprint that can be used to further elucidate the regulation of FA and SSP accumulation processes in soybean seeds.

In summary, we identified 1479 DEGs, 82 DEPs, and 34 DEMs through transcriptomics, proteomics, and metabolomics analyses of dry seeds of R122 and SN14. Among them, DEGs and DEPs responsible for the seed storage profile could be divided into two distinct groups; one group included 18 DEGs and 20 DEPs that were located within the SN14 cultivated soybean genomic background, whereas the other group included 13 DEGs and 4 DEPs that were located within in the R122 substitution regions. GO and KEGG annotations of the two groups showed enrichment in species related to seed storage, indicating a potential link between the DEGs and DEPs and the seed storage profile differences observed between R122 and SN14 (Appendix A).

### 2.3. Integrated Analysis of Transcriptomics, Proteomics, and Metabolomics

To determine if the underlying molecular differences between SN14 and R122 may be responsible for the different seed storage profiles, we examined the expression patterns of DEGs involved in seed-storage-related biological processes. Of these, two located in the substitution region of R122 were genes involved in photosynthesis. Glyma.08G286700 (encoding carbonic anhydrase 2; FC > 3.29) and Glyma.03G068100 (encoding rubisco activase; FC > 5.14) were significantly up-regulated (Figure 5A). We also observed that chlorophyll A-B binding protein (Glyma.09G071400) from the SN14-cultivated soybean genomic background was up-regulated by more than 7-fold, and the abundance of a related DEM (carbamoyl-L-aspartate) increased by 2.79-fold. Overall, DEGs involved in photosynthesis may be important for improving plant performance by impacting the content and quality of the seed storage profile.

Two candidate genes (Glyma.08G233000 and Glyma.03G046100) involved in sugar metabolism were detected from the substituted region and down-regulated by 0.03- and 0.21-fold in R122, respectively (Figure 5B). These genes may participate in shaping the seed storage profile, as β-glucosidase 24 (Glyma.08G233000) functions in the production of monomeric sugars from cellulose-based oligosaccharides and phosphomannomutase family protein (Glyma.03G046100) catalyzes the interconversion of glucose-6-phosphate and glucose-1-phosphate [31,32].

There was a significant difference in fatty acid biosynthesis and lipid metabolism between R122 and SN14, and 5 DEGs, 2 DEPs, and 1 DEM related to these processes were identified from the chromosomal substitution region (Figure 5E).

Two DEGs involved in fatty acid biosynthesis and lipid transfer, Glyma.03G015500 and Glyma.03G040100, were down-regulated by 0.42- and 0.47-fold, respectively. Glyma.03G015500, encoding a thioesterase superfamily protein, plays an important role in the partitioning of FA produced by the de novo pathway. Specifically, thioesterase substrate specificity is a key factor in determining the chain length and the saturation of FAs exported from plastids [33], and lipid transfer protein 1 (Glyma.03G040100) is responsible for the transport of water-insoluble lipids [34]. In this study, we identified two highly up-regulated lipoxygenases, Glyma.08G137000 and A0A0R0KV40, which are important for the degradation of storage lipids and FAs. Long-chain acyl-CoA synthetase (LACS) genes catalyze the addition of CoA, which is the first step of β-oxidation of Fas in peroxisomes [35]. Glyma.11G017900, encoding LACS4, was increased by 8.45-fold, and the abundance of a related DEM (LysoPG 16:0) was decreased by 0.45-fold, supporting the function of LACS4 and possibly explaining our previous results on the total FA content in R122 seeds. I1KXH0 (spermidine synthase 1) is one of the key enzymes involved in the polyamine pathway of lipid metabolism, and its abundance increased by 1.22-fold in R122, which may be relevant. Spermidine synthase (SPDS) uses PUT as a substrate for the biosynthesis of spermidine (SPD), which regulates the biosynthesis of ganoderic acid (GA) [29]. The abundance of seipin 1A, which is located in the SN14-cultivated soybean genomic background, decreased in abundance, providing more evidence for the reduction in total FA content in R122. This is noteworthy as it is a key factor in seed lipid storage processes, including LD biogenesis and TAG accumulation. Remarkably, seipin 1A overexpression in *Arabidopsis* increased seed oil content by up to 10% compared to wild-type seeds [36].

Protein synthesis is a complex process mediated by several molecules (mRNAs, tRNAs, and ribosomes) and organelles, including the endoplasmic reticulum and Golgi bodies [37]. In general, SSP levels are regulated by several biological steps at the transcriptional, translational, and post-translational levels, with post-translational regulation including protein modification, processing and trafficking, and deposition [38]. We identified several DEGs and DEPs from the chromosomal substitution region that exhibited an increase in expression in R122, including the ribosome-related DEG Glyma.08G285000 and a DEP (I1JPV1), as well as the DEGs Glyma.08G279900 and Glyma.03G036700, which are involved in protein post-translational modifications (Figure 5C,D). We also identified Glyma.14G134900, which encodes methionyl-tRNA synthetase and participates in amino acid synthesis, and Glyma.02G043200, a gene related to vacuolar protein sorting. These two DEGs were located in the cultivated soybean genomic background, and integrated analysis showed that the up-regulation of these genes could explain the decrease in L-aspartic acid and the increase in L-malic acid (Appendix A). I1JIA7 and A0A0R0H563, responsible for amino acid biosynthesis, were also identified from the cultivated soybean genomic background, and both genes were up-regulated by 0.38- and 0.36-fold in R122, respectively. I1JIA7 (indole-3-glycerol-phosphate synthase) catalyzes the conversion of 1-(2-carboxyphenylamino)-l-deoxyribulose-5-phosphate to indole-3-glycerolphosphate and plays an important role in tryptophan biosynthesis [39]. Finally, the DEGs Glyma.17G137300 and Glyma.09G155500, which encode the major SSP components Kunitz-type trypsin inhibitor (FC = 6.76) and 2S albumin (FC = 0.35), respectively, together with several proteases and aminopeptidases directly related to SSP levels, were located in the cultivated soybean genomic background (Appendix A).

We identified not only DEGs and DEPs but also 13 DEMs related to the seed storage profile through our integrated analysis, including carbamoyl-L-aspartate and LysoPG 16:0, as mentioned above (Figure 6, Appendix A).

A total of 13 DEMs belonged to five seed storage pathways, including the photosynthesis and energy metabolism pathway, sugar metabolism, lipid metabolism, protein post-translational modification, and storage protein accumulation. Totals of 2 lipid families (LysoPC 18:4 and LysoPG 16:0) and 3 amino acid families [L-arginine, N-α-acetyl-L-ornithine, and aspartic acid (calcium)] showed decreased abundance, whereas 6 amino acid derivatives [L-aspartic acid-O-diglucoside, L-aspartic acid, N-Carbamoyl-L-aspartate, N-Acetyl-L-glycine, L-Lysine-butanoic acid, and S-(2-carboxypropyl)cysteine] and 2 organic acids (L-malic acid and 2-hydroxyisocaproic acid) increased in abundance. Among them, L-malic acid and L-aspartic acid were involved in amino acid biosynthesis, L-aspartic acid-O-diglucoside were involved in protein post-translational modification, exhibiting the highest abundance, whereas S-(2-carboxypropyl) cysteine, N-acetyl-L-glycine, and N-carbamoyl-L-aspartate, which were involved in seed storage accumulation, exhibited the lowest abundance. Finally, we performed qRT-PCR for some genes detected by integrated analysis using the dry seeds of R122 and SN14 (Appendix A) and showed that the relative expression levels of the selected DEGs were consistent with the transcriptomic results. Furthermore, our integrated multi-omics analysis enabled the identification of novel candidate genes that may be related to seed storage processes, including tetratricopeptide repeat (TPR)-like superfamily protein (Glyma.07G097300), telomerase ribonucleoprotein complex-RNA binding domain (Glyma.15G274000), late embryogenesis abundant protein (Glyma.10G017600), ERF9 (Glyma.13G236500), and NAC047 (Glyma.16G151500) (Appendix A). Interestingly, ERF9 and NAC047 are involved in the biogenesis of amino acids, such as N-acetyl-L-glycine and N-carbamoyl-L-aspartate, which may be related to the accumulation of these metabolites in R122. Overall, our integrated multi-omics analyses revealed that R122 possesses a unique molecular fingerprint that can be explored to further elucidate the regulation of FA and SSP accumulation in soybean seed.

### 2.4. Haplotype Analyses of Candidate Gene

In order to narrow the range of candidate genes, we first removed the genes with no difference in expression, and further identified 11 candidate genes through the pathway, including Glyma.02G043200, Glyma.03G015500, Glyma.03G046100, Glyma.03G068100, Glyma.08G279900, Glyma.09G071400, Glyma.09G155500, Glyma.10G162800, Glyma.11G017900, Glyma.14G214800, and Glyma.17G137300. qRT-PCR quantitative verification of these genes was found to be consistent with the trend of transcriptome data, indicating that the transcriptome data were credible (Figure 7A). In order to determine the role of these candidate genes in the process of oil and protein accumulation, we collected 574 resource varieties [40]. We compared the protein content data of these genes in the resource population in 2018 and 2019, and only Glyma.08G279900 has two excellent haplotypes (Figure 7B). Comparing the oil content data, it was found that Glyma.02G043200 had six haplotypes, of which the excellent haplotypes were Hap1 (169), Hap2 (199), Hap3 (55), and Hap4 (101). The number of varieties in the population was more than 5%, which was the excellent haplotype. There was a significant difference in oil content between Hap and Hap3, and Hap4 and Hap2, but there was no significant difference between Hap1, Hap3, and Hap4. There may be a close relationship between Hap2 and soybean oil content (Figure 7C). Glyma.03G068100, Glyma.08G279900, and Glyma.11G017900 have two haplotypes, and there are significant differences in oil content (Figure 7D–F). Glyma.14G214800 has seven haplotypes, of which the excellent haplotypes are Hap1 (428), Hap2 (53), and Hap3 (47). There is a significant difference between Hap1, Hap2, and Hap3, and there is no significant difference between Hap1 and Hap2 (Figure 7G). Glyma.17G137300 has 11 haplotypes, and the excellent haplotypes are Hap1 (304 copies), Hap2 (100 copies), Hap5 (39 copies), and Hap6 (39 copies) (Figure 7H). Therefore, these haplotype results support that the candidate genes Glyma.02G043200, Glyma.03G068100, Glyma.08G279900, Glyma.11G017900, and Glyma.17G137300 may have haplotypes affecting oil content, and the haplotype of Glyma.08G279900 may be closely related to protein content.

## 3. Discussion

Due to its high protein content, soybean is a good source of plant protein, and its production and consumption worldwide are increasing annually. The main products derived from soybean include soy milk, soy oil, soybeans, soy sauce, and tofu [41]. Soybean products may be effective for the treatment of several diseases, including cardiovascular disease and cancer, and may be particularly helpful for menopausal women in the prevention of osteoporosis. Furthermore, unsaturated FAs abundant in soybean oil are popular products that lower serum cholesterol levels and protect against heart disease [6]. An ever-increasing demand for soybean products calls for informed breeding measures to produce soybean varieties with high oil and protein contents, and thus it is important to have a thorough understanding of the molecular mechanisms for the synthesis and accumulation of seed oil and seed storage protein.

Individuals in a CSSL population can be regarded as “artificial mutants” that carry different substituted chromosomal fragments donated by a wild plant. Thus, they are widely applied in the field of agricultural genetics because the agronomic traits that differ from the recurrent parent can be assessed at the genetic level for causality [28]. A detailed analysis of substituted chromosome segments from a CSSL is a prerequisite for elucidating why the CSSL has a different trait from the recurrent parent, and thus could enable the discovery of novel genes or regulatory elements responsible for favorable agronomic traits. Many agronomic traits, including seed storage profiles, involve complex crosstalk between different biological layers (genome, proteome, and metabolome), and integrated multi-omics analyses can assess their underlying molecular context [42]. To obtain a more comprehensive molecular picture of CSSL with seed storage profiles different from the recurrent parent, in this study, we performed transcriptomics, proteomics, and metabolomics analyses followed by integrated analysis. Integrated analysis enabled us to identify novel genes putatively involved in seed storage profile networks that were not previously related to seed storage profiles, which enabled us to relate DEG and DEP information to the metabolic fingerprint of the CSSL.

We identified 63 DEGs, 4 DEPs, and 1 DEM from the substitution region, which is involved in photosynthesis and energy metabolism, sugar metabolism, ribosomal protein-related pathways, protein post-translational modifications, and lipid metabolism. Carbonic anhydrase 2 (CA2) expression may explain the differences in the seed storage profile of R122 because when the cytosolic carbonic anhydrase from Flaveria bidentis was overexpressed ectopically in *Arabidopsis thaliana*, the total protein content and biosynthesis of several amino acids were enhanced [43]. Rubisco activase (RCA) enhances rubisco activation by facilitating the removal of inhibitors in an ATP-dependent manner, and its overexpression can improve the photosynthetic rate and growth performance of rice, increasing the dry weight of the overall plant by 26%, even under high-temperature conditions [44]. A previous study reported that the overexpression of phosphomannomutase (Glyma.03G046100) also significantly increased starch, glucose, and fructose levels and decreased the sucrose level in sweet potato fruits [32]. Interestingly, several of the identified DEGs and DEPs are responsible for lipid metabolism, including FA biosynthesis, lipid transfer, TAG storage, and FA degradation, which are processes that could influence seed oil content in R122. Although the molecular mechanism is uncharacterized, the identification of spermidine synthase related to TAG storage was identified through our integrated analysis, and it may be a unique gene to pursue soybean seed molecular breeding. In addition, the decreased abundance of the seed storage-related DEM LysoPG 16:0 provides a context for the low oil content in R122. GO and KEGG annotations showed that DEGs and DEPs responsible for SSP belonged to processes related to photosynthesis, ribosomal protein-related pathways, and protein post-translational modifications. Several novel candidate DEGs and DEPs that may be related to SSP accumulation, as well as amino acid synthesis, were identified through our integrated analysis and detected in the cultivated soybean genomic background, including ERF9 (Glyma.13G236500) and NAC047 (Glyma.16G151500) (Appendix A). As for seed storage-related DEMs identified through our integrated analysis, the contents of L-arginine, LysoPG 16:0, and LysoPC 18:4 were decreased, whereas those of other amino acid derivatives and organic acids such as L-malic acid were increased. While the molecular mechanism underlying the amino acid profile of R122 is uncharacterized, future studies may reveal an important regulatory framework that can be leveraged for breeding higher-quality seeds.

In summary, we propose a regulatory network to explain why the FA content was lower than the protein content in R122 (Figure 8).

CA2, which catalyzes the interconversion of CO_2_ and HCO^3−^, was up-regulated in R122, and this process increased photosynthetic capacity and energy supply by enhancing the flux of carboxylic acid to the TCA cycle and CO_2_ to the Calvin–Benson cycle [43]. RCA activates rubisco, which is one of the key enzymes in the Calvin–Benson cycle, to improve the rate of photosynthesis [43]. This increased photosynthetic capacity can help drive the expression and activity of the DEGs and DEPs related to protein biosynthesis at the translational and post-translational levels. This phenomenon can lead to the up-regulation of these proteins, including ribosomal proteins, HSP proteins involved in protein processing in the endoplasmic reticulum, and proteins involved in ubiquitination, which could ultimately enhance the accumulation of SSPs in seeds [38]. On the other hand, the up-regulation of LOX and LACS4 can result in the degradation of FAs and an overall decrease in the FA content of seed [35].

In this study, several candidate genes related to photosynthesis and storage sink compounds were identified through multi-omics integrated analysis. Interestingly, the genes that were located in the substitution region have not been reported to be related to seed storage profiles. Although we speculate that the candidate genes related to photosynthesis are more related to the early-maturity and mid-maturity stages of seed development, we speculate that photosynthesis is the earliest stage of plant synthesis of storage substances, and there may be a more complex transformation process, which needs further discussion and research. These putative seed storage genes require further functional characterization, and our results contribute to understanding the regulatory network of seed storage genes and to designing future studies to improve seed quality.

## 4. Materials and Methods

### 4.1. Plant Material

The CSSL population previously constructed by our lab was planted and grown at Xiangyang Farm, Harbin, China (45.75° N, 126.53° E), in May of 2019 and 2020 [28]. A random block design was adopted, and each line was planted in a row in triplicate. The rows were 5 m long and 65 cm wide. Each single-row plot contained approximately 80 plants that were 6 cm apart. To screen for CSSLs with significantly different oil and protein contents from the recurrent line (SN14), basic phenotypes (flower color, leaf shape, and growth habit), growth stage, flowering time, yield-related traits (seed size, seed length and width, 100-seed weight, and seed weight per plant), and plant height were investigated [45]. If the CSSLs showed significant differences from the SN14 line in terms of these characteristics, they were excluded from follow-up analyses of oil and protein contents to limit the effects of growth and developmental-related phenotypes on seed quality. Three replicate samples from the screened CSSLs were used to perform FA and SSP content analyses. R122, which exhibited significant FA and SSP differences from the SN14 line, was selected from the CSSL population through FA and SSP content analyses (Appendix A).

### 4.2. Fatty Acid Profile and Seed Storage Protein Content Analyses

The FA profile and SSP content analyses were performed as previously described [46,47]. In brief, 5 mg of soybean powder from dried seeds was used for FA profile analysis employing an Agilent 7890B GS system. Soybean seed powder was mixed with 100 µg of heptadecanoic acid and added to the extraction solution (2.5% [*v*/*v*] H_2_SO_4_ in CH3OH). The sample solution was incubated at 85 °C for 1 h and centrifuged at 5975× *g* for 10 min, followed by the addition of 150 µL of 0.9% (*w*/*v*) NaCl and 700 µL of hexane. After air drying, the residual FA methyl esters were dissolved in 400 µL of ethyl acetate for GC analysis. An amount of 50 mg of soybean dried powder which was filtered through a 60-mesh screen was used for SSP content analysis employing an NDA702 Dumas analyzer (VELP, Usmate Velate, Italy). The oil and protein contents (%) of samples were averaged over three replications for each sample. For sodium dodecyl sulfate polyacrylamide gel electrophoresis (SDS-PAGE) analysis, R122 and SN14 soybean seed powders were passed through a 60-mesh screen, suspended in 200 µL of lysis buffer (4% SDS, 100 mM DTT, and 150 mM Tris-HCl [pH 8.0]) on ice, and then quantified with a bicinchoninic acid protein assay kit (Bio-Rad, Hercules, CA, USA). About 15 µg of protein from each sample was resolved by 12% SDS-PAGE in 5:1 (*v*/*v*) 5× stacking buffer with Coomassie Brilliant blue G-250 (Sigma, St. Louis, MA, USA). Statistical significance for FA and SSP contents was calculated by one-way ANOVA or Student’s *t*-test (*p* < 0.05) using SPSS 17.0 and Microsoft Excel 2016, respectively.

### 4.3. RNA-Seq Analysis

Three replicate samples for RNA-seq were selected from the dry seeds of R122 and SN14. Total RNA was extracted and purified from tissue ground in liquid nitrogen using TRIzol reagent (Invitrogen, Carlsbad, CA, USA). After the concentration and purity of RNA were assessed via a NanoDrop 2000C spectrophotometer (Thermo Scientific, Waltham, MA, USA), mRNA purified from 50 µg of total RNA was used for RNA-seq. High-throughput sequencing was performed on an Illumina NovaSeq 6000 system to generate 150 bp paired-end reads. Analysis of the resulting RNA-seq data, including the mapping of raw reads to the reference genome, counting of reads, and calculation of fragments per kilobase of exon per million fragments mapped, was performed as described previously [45]. The soybean reference genome (Wm82.a2.v1) and annotation dataset were downloaded from the Phytozome database, and raw sequencing data were deposited in the National Center for Biotechnology Information database with the bio-project accession number PRJNA939828 (https://dataview.ncbi.nlm.nih.gov/object/PRJNA939828?reviewer=ogtugl1r9vu0p6j8fu1lqe1h22 (accessed on 1 March 2023)). To compare SN14 and R122 transcript abundances, the threshold for identifying differentially expressed genes (DEGs) was set at log2 FC >1 or <−1 (Student’s *t*-test: *p* < 0.05).

### 4.4. Proteomics Analysis

Three replicate samples for proteomics were selected from the dry seeds of R122 and SN14. A Q-Exactive HF-X mass spectrometer coupled to an Easy nLC 1200 liquid chromatograph (Thermo Fisher Scientific, Waltham, MA, USA) was used for liquid chromatography–tandem mass spectrometry (LC–MS/MS) analysis, and data-dependent acquisition profile analysis was performed on the Q-Exactive HF-X mass spectrometer for 90 min with the detection parameters described previously [48]. The LC–MS/MS raw data were imported into Proteome Discoverer (v2.4; Thermo Fisher Scientific, Waltham, MA, USA) for protein identification against the file max_Soybean_3847_85057_202109.fasta in the Uniprot_Glycine database (downloaded from https://www.uniprot.org/taxonomy/3847 (accessed on 1 March 2023), with 85,057 protein sequences, on 09/2021), and the results and analysis parameters are shown in Appendix A. To compare R122 and SN14, the threshold for differentially expressed proteins (DEPs) was set at FC >1.20 or < 0.83 (Student’s *t*-test: *p* < 0.05), and DEPs were grouped based on protein levels as described previously. Protein sequences were subsequently annotated against UniProtKB/Swiss-Prot, Kyoto Encyclopedia of Genes and Genomes (KEGG), and Gene Ontology (GO) databases. GO term and KEGG pathway analyses were performed using Fisher’s exact test with the false discovery rate correction.

### 4.5. Metabolomics Analysis

Three replicate samples for metabolomics were selected from the dry seeds of R122 and SN14. Samples were crushed using a mixer mill (Retsch-MM 400, Haan, Germany) and lyophilized in a vacuum freeze-dryer (Scientz-100F, Beijing, China). About 100 mg of lyophilized powder was dissolved in 1.2 mL of 70% methanol (Invitrogen, Carlsbad, CA, USA), and the samples were vortexed before storing at 4 °C overnight. After centrifugation at 12,000 rpm for 10 min, the extracts were filtered (SCAA-104, 0.22-μm pore size; ANPEL, Shanghai, China) and analyzed using a UPLC-ESI-MS/MS system (UPLC, SHIMADZU Nexera X2; MS, Applied Biosystems 4500 Q TRAP, Tokio, Japan) equipped with an Agilent SB-C18 column (1.8 µm, 2.1 mm × 100 mm). Linear ion trap (LIT) and triple quadrupole (QQQ) scans were acquired on a Q-TRAP mass spectrometer and an AB4500Q TRAP UPLC/MS/MS system. The ESI source operation parameters were as follows: ion source, turbo spray; source temperature, 550 °C; ion spray voltage (IS), 5500 V (positive ion mode)/4500 V (negative ion mode); ion source gas I (GSI), gas II (GSII), and curtain gas (CUR) were set to 50, 60, and 25.0 psi, respectively; and the collision-activated dissociation (CAD) was high [49]. Differentially expressed metabolites (DEMs) were screened out using the criteria of FC >2 and <0.5, and variable importance (VIP) > 1.

### 4.6. qRT−PCR Validation of Candidate Genes

RNA was extracted from tissue homogenized using TRIzol reagent. After the concentration and purity were assessed with a NanoDrop 2000C spectrophotometer (Thermo Scientific, Waltham, MA, USA), RNA was synthesized into cDNA using a HiScript II qRT SuperMix (+gDNA wiper) kit. qRT-PCR was performed on cDNA using the 2× ChamQ Universal SYBR qPCR Master Mix (Vazyme Biotech, Nanjin, China) in a Light Cycler 480 system (Roche Diagnostics, Switzerland). qRT-PCR primers for the candidate genes were designed using Premier 5.0 software and are shown in Appendix A. GmActin4 was used as an internal reference, and the relative expression levels were estimated as the average of three biological replicates calculated by the 2–ΔΔCt method [8]. Statistical significance for relative expression level were calculated by one-way ANOVA or Student’s *t*-test (*p* < 0.05) using SPSS 17.0 and Microsoft Excel 2016, respectively.

### 4.7. Statistical Analysis

Statistical significance for the comparative analysis of CSSL phenotypes and for expression level analysis of genes, proteins, and metabolites was calculated by one-way ANOVA or Student’s *t*-test (*p* < 0.05) using SPSS 17.0 and Microsoft Excel 2016, respectively. The omics raw data were analyzed by R language (https://www.r-project.org/) accessed on 1 March 2023.

### 4.8. Haplotype Analysis

A total of 500 germplasm resources with rich genetic variation were used for haplotype analysis. Local BLAST analysis was performed on the candidate gene sequence information and re-sequencing genome sequence information of 500 soybean germplasm resources to obtain SNP information of candidate genes in the germplasm resource population. In this study, DNAsp5.0 software was used to analyze the haplotype distribution of candidate gene SNP sequences in germplasm resource populations, and excellent haplotypes (varieties with haplotypes exceeding 5%) were screened.

## Figures and Tables

**Figure 1 ijms-25-05614-f001:**
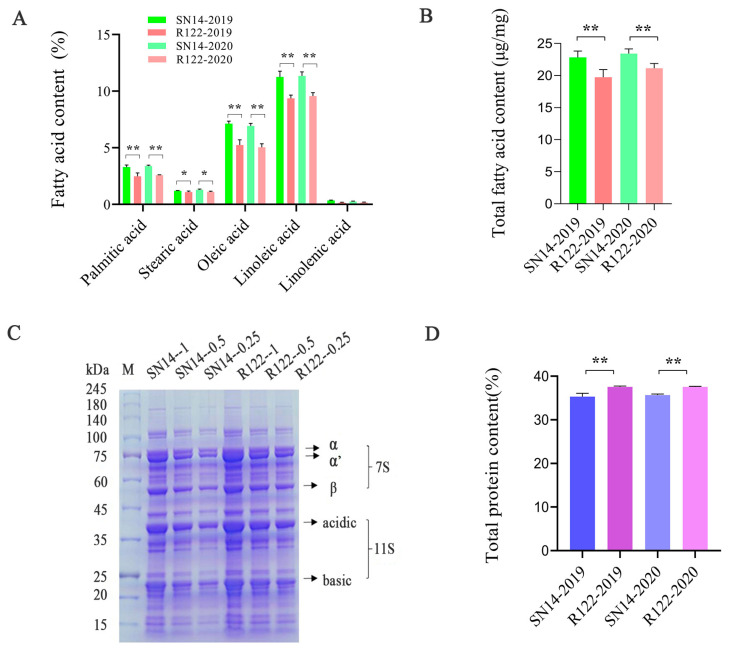
Comparative analysis of total protein and fatty acids between R122 and SN14. (**A**) Analysis of fatty acid profile. (**B**) Analysis of total fatty acid content. (**C**) Analysis of sodium dodecyl sulfate polyacrylamide gel electrophoresis (SDS-PAGE) protein profile by concentration gradients with a dilution series of 1×, 0.5×, and 0.25×. (**D**) Analysis of total protein content. Student’s *t*-test: ** *p* < 0.01; * *p* < 0.05.

**Figure 2 ijms-25-05614-f002:**
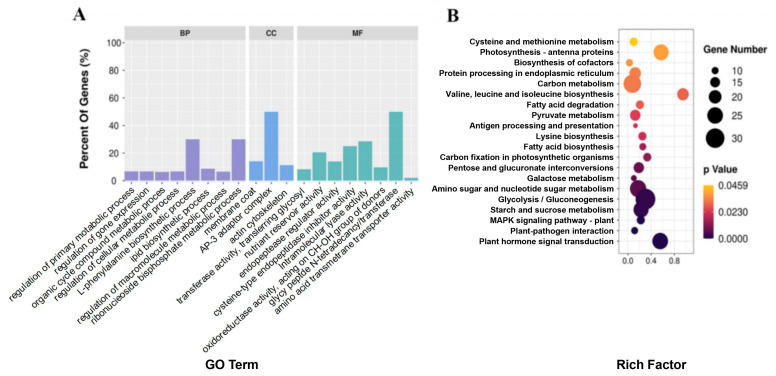
Comparative transcriptomics between R122 and SN14. (**A**) Gene Ontology (GO) analysis of total differentially expressed genes (DEGs). Three categories of GO analysis: biological process (BP), cell component (CC), molecular function (MF). (**B**) Kyoto Encyclopedia of Genes and Genomes (KEGG) analysis of total DEGs.

**Figure 3 ijms-25-05614-f003:**
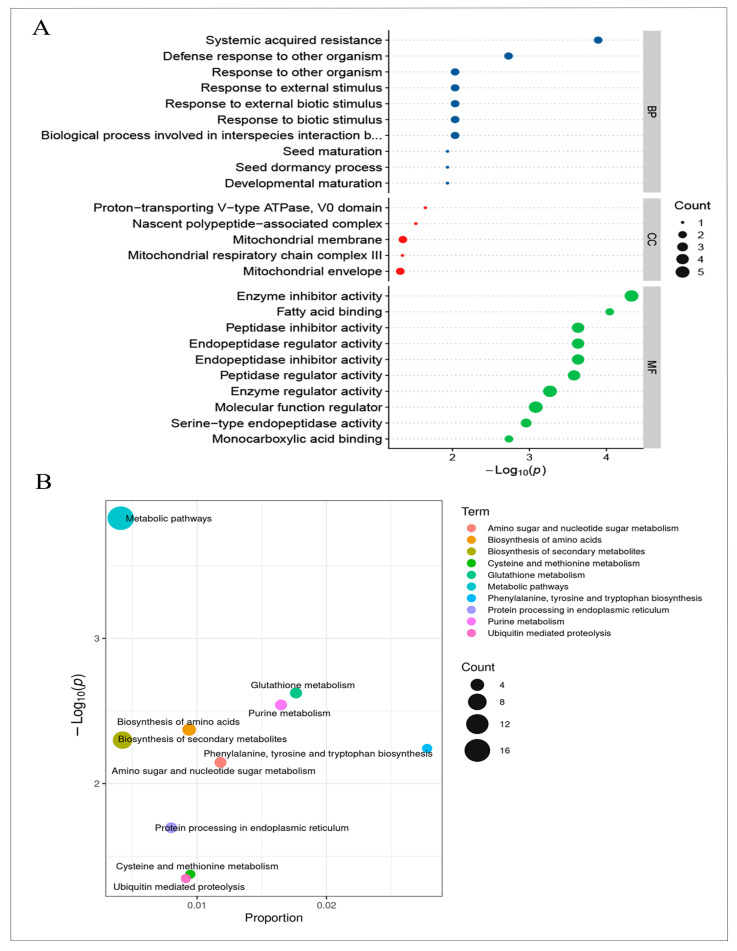
Comparative proteomics between R122 and SN14. (**A**) GO analysis of total differentially expressed proteins (DEPs).The blue, red and green colors in the diagram indicate biological processes, cellular components and molecular functions, respectively. (**B**) KEGG analysis of total DEPs.

**Figure 4 ijms-25-05614-f004:**
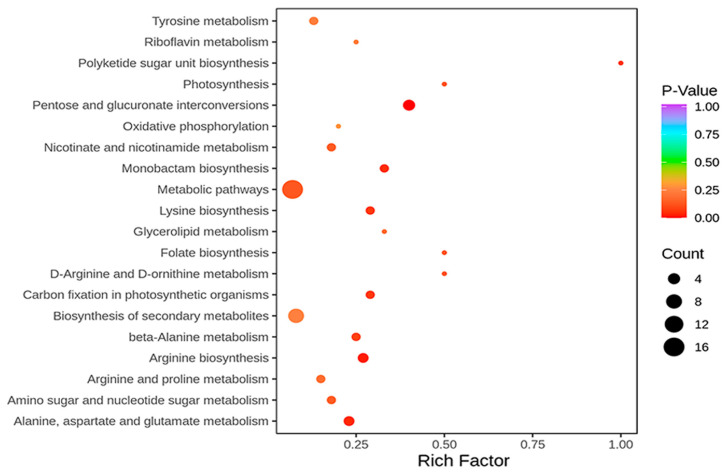
KEGG annotation of total differentially expressed metabolites (DEMs) for comparative metabolomics between R122 and SN14.

**Figure 5 ijms-25-05614-f005:**
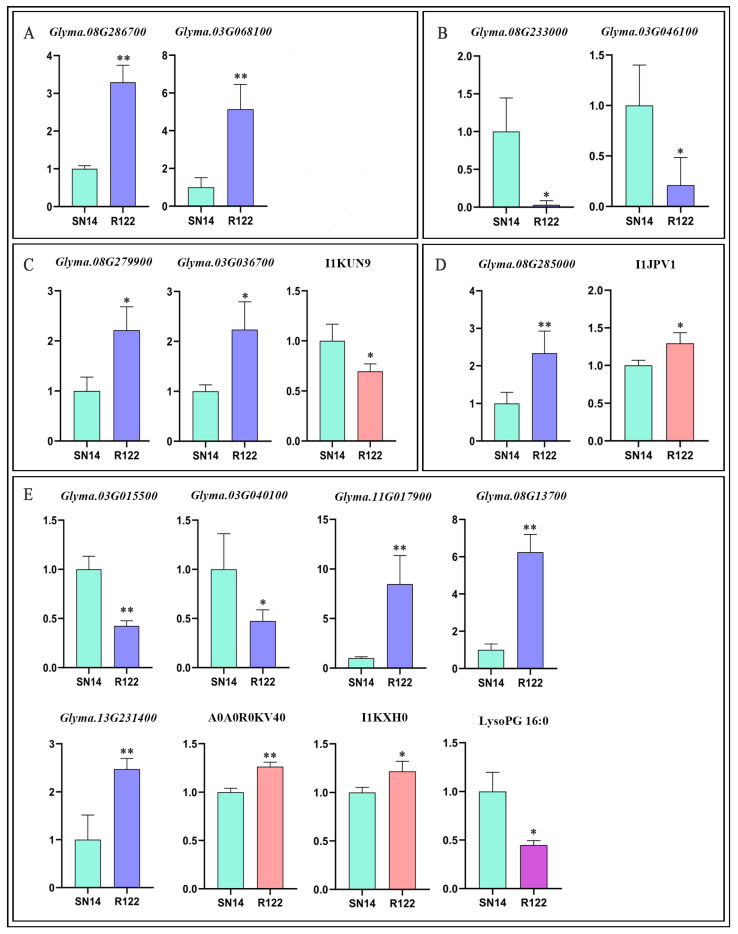
Integrated analysis of DEGs, DEPs, and DEMs located in the substituted region. (**A**) Photosynthesis and energy metabolism pathway. (**B**) Sugar metabolism. (**C**) Protein post-translational modification. (**D**) Ribosomal-protein-related pathway. (**E**) Lipid metabolism. Student’s *t*-test: ** *p* < 0.01; * *p* < 0.05.

**Figure 6 ijms-25-05614-f006:**
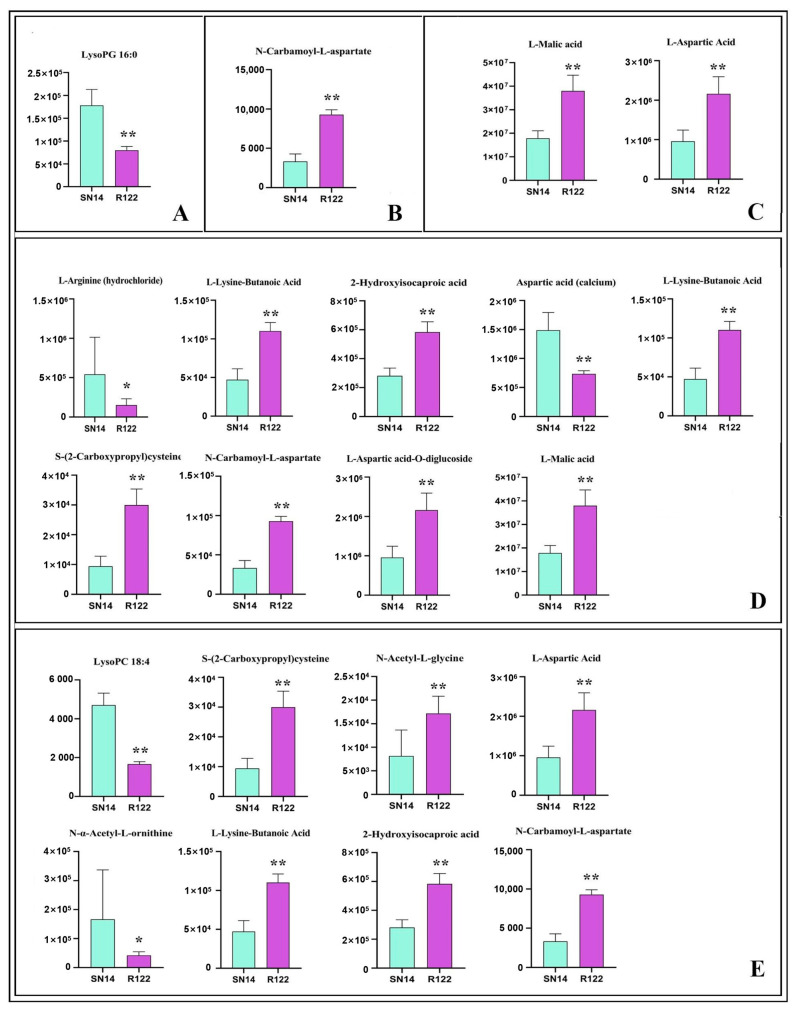
Comparative analysis of DEMs related to seed storage between R122 and SN14. (**A**) Lipid metabolism. (**B**) Photosynthesis and energy metabolism pathway. (**C**) Amino acid biosynthesis. (**D**) Protein post-translational modification. (**E**) Storage protein accumulation. Student’s *t*-test: ** *p* < 0.01; * *p* < 0.05.

**Figure 7 ijms-25-05614-f007:**
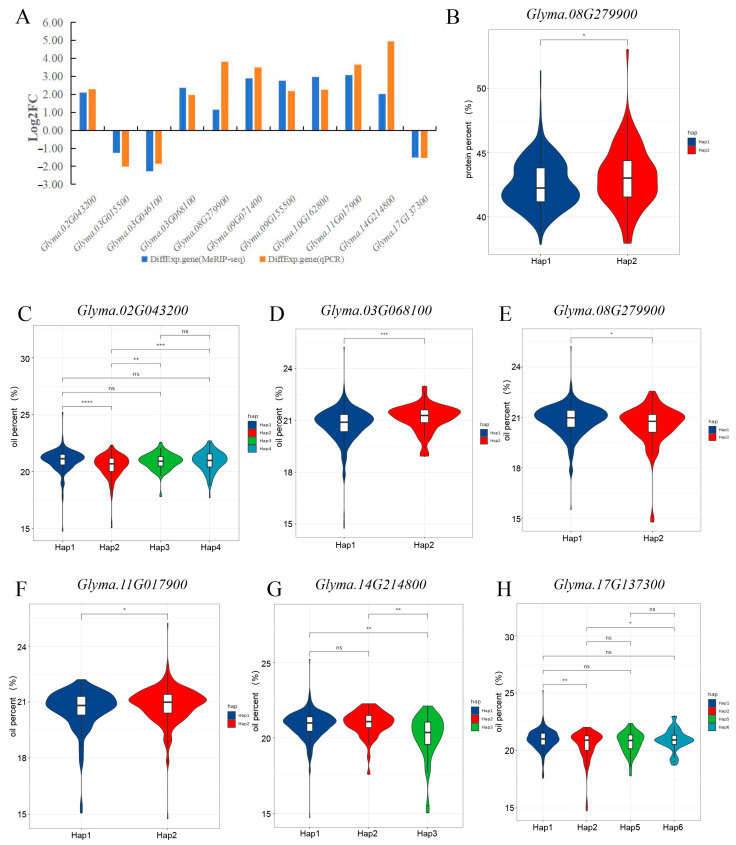
Candidate gene expression and haplotype analysis. (**A**) Candidate gene qRT−PCR statistical diagram. Blue is the transcriptome data and yellow is the qRT−PCR data. (**B**) Haplotype of candidate gene protein content. (**C**–**H**) Candidate gene oil content haplotype. The number represents the degree of significance. * indicates that the *p* value is less than 0.05, ** indicates that the *p* value is less than 0.01, *** indicates that the *p* value is less than 0.001, a *p* value less than 0.0001 is represented by ****, and ns indicates that there is no significant relationship.

**Figure 8 ijms-25-05614-f008:**
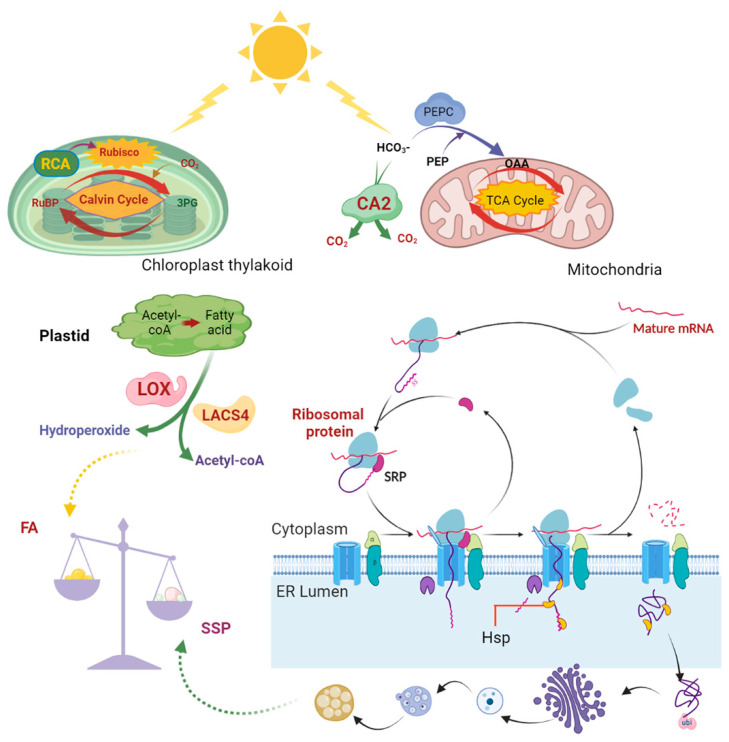
Proposed regulation network for the accumulation of fatty acids and seed storage proteins in soybean. Abbreviations: CA2—carbonic anhydrase2; RCA—rubisco activase; LOX—lipoxygenase; LACS4—long-chain acyl−CoA synthetase4; Hsp—heat shock protein; ubi—ubiquitination; RuBP—ribulose bisphosphate; 3PG—3-phosphoglycerate; OAA—oxaloacetic acid; PEP—phosphoenolpyruvate; PEPC—phosphoenol pyruvate carboxylase; TCA cycle—tricarboxylic acid cycle; SRP—signal recognition particle.

## Data Availability

Data is contained within the article.

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
