# Peer review of "Multi-Omics Analysis of a Chromosome Segment Substitution Line Reveals a New Regulation Network for Soybean Seed Storage Profile"

_ijms, 2024, doi:10.3390/ijms25115614_

Round 1

Reviewer 1 Report

Comments and Suggestions for Authors

The manuscript with the title “Multi-omics analysis of a chromosome segment substitution line reveal a new regulation network for soybean seed storage profile” chose chromosome segment substitution line (CSSL) (line characterized by high-protein and low-oil content) to explore the underlying effect of donor introgression on seed storage through multi-omics analysis. The work has good degree of novelty and is of current interest because soy is a multipurpose crop cultivated on large areas.

The manuscript is written in a logical sequence, methods used are described in detailed and represent recognized approaches – hence there is a good degree of credibility for the results obtained.

The paragraph that expresses the aim and objectives shall be distinctively presented at the end of the introduction section, and I suggest these shall not contain results but only aim and list of objectives.

The authors reached their objectives for this research.

The references are on the topic.

Best regards.

Comments on the Quality of English Language

minor English style and grammar improvements are recommended

Author Response

Thank you for your suggestions, we have modified the end of the preface.

Reviewer 2 Report

Comments and Suggestions for Authors

"Multi-omics analysis of a chromosome segment substitution line reveal a new regulatory network for soybean seed storage profile" manuscript has been well prepared and the comments are mostly editorial.

I recommend first of all, taking into account the opinion of a reviewer specializing in this area.

Lines 8-9 - typo

Lines 32-33 - The statistical data concerns soybean  in general or only used for human consumption 

Line 63 - All Latin name should be written Italic

Line 77 - Comments the same as above

Line 81 editorial mistake

Line 108-111 The sum of fatty acids presented in the table: palmitic acid, stearic acid, oleic acid, linoleic acid, and linolenic acid contents ... were 0.80 to 0.82%, 0.08 to 0.20%, 1.89 to 1.93%, 1.79 to 1.89%, and 0.09 to 0.18 % is different than that authors presented by the authors (not 3.09 but 4.65% and not 2.29 but 5.02%). Please correct or add other FA.

Line 123 - No data (tables and figures) in the supplement.

Please explain what does mean BP, CC and MF on Gene ontology (Fig. 2A).

Fig. 2A. The sum of the genes is greater than 100. The information requires explanation

Fig. 2A. Too small font in the description under the "X" axis

Lines 259 and 264  Latin name - please write Italic.

Lines 257-260 Why the spermidine synthase results from the study on TAG storage in Drosophila were cited instead of synthesis in plants?

Fig. 7B - typo (axis Y description)

Please explain why the discussion was partly carried out in the description of the results (Chapter 2) and partly in Chapter 3

Line 387-388 - Scientific species name - please write in Italic

Lines 423 and many others - incorrect writing of chemical formulas f.ex. CO2 should be CO2

Line 444 - typo

Line 445 - no sowing date of soybean

Author Response

Lines 8-9 - typo

Thank you for your suggestions, we have modified it.

Lines 32-33 - The statistical data concerns soybean  in general or only used for human consumption

Thank you for your suggestions, we have modified it.

Line 63 - All Latin name should be written Italic

Thank you for your suggestions, we have modified it.

Line 77 - Comments the same as above

Thank you for your suggestions, we have modified it.

Line 81 editorial mistake

Thank you for your suggestions, we have modified it.

Line 108-111 The sum of fatty acids presented in the table: palmitic acid, stearic acid, oleic acid, linoleic acid, and linolenic acid contents ... were 0.80 to 0.82%, 0.08 to 0.20%, 1.89 to 1.93%, 1.79 to 1.89%, and 0.09 to 0.18 % is different than that authors presented by the authors (not 3.09 but 4.65% and not 2.29 but 5.02%). Please correct or add other FA.

Thank you for your suggestions, we apologise for this wrong calculation and modified it.

Line 123 - No data (tables and figures) in the supplement.

Thank you for your suggestions, we re-upload the supplementary figures and supplementary tables.

Please explain what does mean BP, CC and MF on Gene ontology (Fig. 2A).

Thank you for your suggestions, we have added relevant content to the figure notes, BP, CC and MF stand for Biological Processes, Cellular Components and Molecular Functions.

Fig. 2A. The sum of the genes is greater than 100. The information requires explanation

The interaction between genes is complex, and the same gene may also play different functions in different signaling pathways and functional ontologies, so some classifications need to be made, so the sum of the genes is greater than 100.

Fig. 2A. Too small font in the description under the "X" axis

Thank you for your suggestions, we have made changes to enlarge the text.

Lines 259 and 264  Latin name - please write Italic.

Thank you for your suggestions, we have modified it.

Lines 257-260 Why the spermidine synthase results from the study on TAG storage in Drosophila were cited instead of synthesis in plants?

Thank you for your suggestions, we have modified it.

Fig. 7B - typo (axis Y description)

Thank you for your suggestions, we have switched to plant-related research.

Please explain why the discussion was partly carried out in the description of the results (Chapter 2) and partly in Chapter 3

Thank you for your suggestions, we added a detailed description in the multi-omics analysis part, and we compared our results to make them more convincing. So we have a simple discussion.

Line 387-388 - Scientific species name - please write in Italic

Thank you for your suggestions, we have modified it.

Lines 423 and many others - incorrect writing of chemical formulas f.ex. CO2 should be CO2

Thank you for your suggestions, we have modified it.

Line 444 - typo

Thank you for your suggestions, we have modified it.

Line 445 - no sowing date of soybean

Thank you for your suggestions, we have added relevant content to the material method.

Reviewer 3 Report

Comments and Suggestions for Authors

The manuscript entitled ”Multi-omics analysis of a chromosome segment substitution line reveal a new regulation network for soybean seed storage profile” deals with regulatory mechanisms to determine protein and lipid accumulation in soybean seeds comparing 2 lines. However, I don't agree the experimental design to analyze the gene expression and metabolite concentraton in dry seeds. The analysis of the protein and lipid should be done using dry seeds, but the gene expression and metabolite should be analyzed in immature seeds during the accumulation of lipid and protein.

Moreover, the authors indicated that photosynthesis is related to the accumulation of lipids and protein, however, soybean seeds import sucrose from leaves, and the sugar and amino acids metabolism in seeds should be regulated to determine lipid and protein concentration in seeds.

Author Response

Thank you for your suggestions, the process of seed formation in soybean can be divided into early-seed maturity stage, mid-seed maturity stage, late-seed maturity stage and dry seed stage, we considered that the dry seed as the last period of seed formation, there may be some differences in the expression of protein and lipid related genes and metabolite changes, so we chose to carry out a combined transcriptome and metabolome and proteome analysis on the dry seeds of two different varieties. Soybean photosynthesis results in an increase in organic matter, which in turn increases the proteins and lipids in the seed. Amino acids are the basic units that make up proteins. Sucrose produced from the leaves supports the energy required for plant physiological activities, and sucrose can be hydrolysed to glucose and fructose, both of which can be enzymatically cleaved to acetyl coenzyme A, which enters the fatty acid synthesis pathway to synthesise lipids. Therefore, we suppose that photosynthesis is related to the accumulation of lipids and proteins.

Round 2

Reviewer 2 Report

Comments and Suggestions for Authors

No more comments. Authors accept all suggestion. Manuscript is suitable for publication.

Author Response

Thank you very much for your efforts for our manuscripts.

Reviewer 3 Report

Comments and Suggestions for Authors

I understand the authors` reply to my review, however, there is no revision for these points.

1) Please add the explanation "the process of seed formation in soybean can be divided into early-seed maturity stage, mid-seed maturity stage, late-seed maturity stage and dry seed stage, we considered that the dry seed as the last period of seed formation" in the introduction. Please explain the characteristics of four stages expecially dry seed stage.

2) Concerning the relationship between photosynthesis and seed storage compounds suggested in Figure 8 is mainly related to the early-, and mid-maturity stages. Also, the characteristics of gene expression and metabolite concentration in seeds are not directly related to the photosynthetic performance in leaves.

Comments on the Quality of English Language

English is good.

Author Response

1) Please add the explanation "the process of seed formation in soybean can be divided into early-seed maturity stage, mid-seed maturity stage, late-seed maturity stage and dry seed stage, we considered that the dry seed as the last period of seed formation" in the introduction. Please explain the characteristics of four stages expecially dry seed stage.

Thank you for your suggestion, I have added the relevant content in the introduction part.

2) Concerning the relationship between photosynthesis and seed storage compounds suggested in Figure 8 is mainly related to the early-, and mid-maturity stages. Also, the characteristics of gene expression and metabolite concentration in seeds are not directly related to the photosynthetic performance in leaves.

Thank you for your question, we agree with you and added the discussion. Figure 8 is only a proposed regulation network. The result of soybean photosynthesis is the increase of organic matter, which is the initial stage of soybean seed storage, mainly related to the early-maturity and middle-maturity stage. The development of soybean seeds from the early-maturity stage to the dry seed stage is also the process of storage material accumulation. It is a more complex process from the beginning of photosynthesis of soybean to the formation of storage substances in dry seed stage, which needs further research and discussion.

Round 3

Reviewer 3 Report

Comments and Suggestions for Authors

In the first part of your revision, please add the validity and rightfulness to study matured seeds for investigating oil and protein accumulation processes. I think especially mRNAs remained in the dry seeds were only residual materials and did not have physiological meanings.

Line 40-41: The sentence "The lipid and protein differences among different varieties were the largest and relatively stable." is difficult to understand. Do you mean "The differences in the lipid and protein concentrations among different varieties were large but those in each variety were relatively consistent."?

Comments on the Quality of English Language

English is good.

Author Response

In the first part of your revision, please add the validity and rightfulness to study matured seeds for investigating oil and protein accumulation processes. I think especially mRNAs remained in the dry seeds were only residual materials and did not have physiological meanings.

Thank you for your suggestions. Matured seeds is the final steps for soybean oil and protein accumulation. The expression level of mRNA in dry seeds is very low, but many of the genes expressed in this part are related to the storage material of seeds. Compared with other legumes, soybean seeds are excellent protein accumulators, and their protein content is as high as 50 % of dry weight. Among them, 80-90 % are two storage globulins: soybean globulin (11S globulin) and β-conglycinin (7S globulin) [1,2]. β-conglycinin has a trimer structure composed of α, α ′ and β subunits, which can be stored in seeds with natural proteins for a long time without losing viability [3].

[1] Velasquez M T, Bhathena S J. Role of dietary soy protein in obesity[J]. International journal of medical sciences, 2007, 4(2): 72.

[2] Cam A, de Mejia E G. Role of dietary proteins and peptides in cardiovascular disease[J]. Molecular nutrition & food research, 2012, 56(1): 53-66.

[3]Vianna G R, Cunha N B, Rech E L. Soybean seed protein storage vacuoles for expression of recombinant molecules[J]. Current Opinion in Plant Biology, 2023, 71: 102331.

Line 40-41: The sentence "The lipid and protein differences among different varieties were the largest and relatively stable." is difficult to understand. Do you mean "The differences in the lipid and protein concentrations among different varieties were large but those in each variety were relatively consistent."?

Thank you for your suggestion, we have made relevant modifications in the manuscript.
